analytical chemistry/chemical biology

Bruton's tyrosine kinase, spebrutinib, metabolic stability, intrinsic clearance, LC-MS/MS

**Author for correspondence:**
Mohamed W. Attwa
e-mail: mzeidan@ksu.edu.sa

This article has been edited by the Royal Society of Chemistry, including the commissioning, peer review process and editorial aspects up to the point of acceptance.

# A highly sensitive LC-MS/MS method to determine novel Bruton's tyrosine kinase inhibitor spebrutinib: application to metabolic stability evaluation

Ali S. Abdelhameed[1], Mohamed W. Attwa[1,2], Nasser S. Al-Shaklia[1] and Adnan A. Kadi[1]

[1]Department of Pharmaceutical Chemistry, College of Pharmacy, King Saud University, PO Box 2457, Riyadh 11451, Saudi Arabia
[2]Students' University Hospital, Mansoura University, Mansoura 35516, Egypt

MWA, 0000-0002-1147-4960

Spebrutinib (SBT) is a Bruton's tyrosine kinase inhibitor. SBT is currently in phase II and phase I clinical trials for the management of rheumatoid arthritis and chronic lymphocytic leukaemia, respectively. We developed and validated a liquid chromatography tandem mass spectrometry analytical method to quantify SBT and investigate its metabolic stability. SBT and the naquotinib as internal standard were isocratically eluted on a C18 column. The linearity of the developed method is 5–500 ng ml$^{-1}$ ($r^2 \geq 0.9999$) in the human liver microsomes (HLMs) matrix. Good sensitivity was approved by the very low limit of detection (0.39 ng ml$^{-1}$). Inter- and intra-assay accuracy values of −1.41 to 12.44 and precision values of 0.71% to 4.78%, were obtained. SBT was found to have an *in vitro* half-life (82.52 min) and intrinsic clearance (8.4 µl min$^{-1}$ mg$^{-1}$) as computed following its incubation with HLMs. The latter finding, hypothesize that SBT could be slowly excreted from the body unlike other related tyrosine kinase inhibitors. So, drug plasma level and kidney function should be monitored because of potential bioaccumulation. To the best of our knowledge, this is considered the first analytical method for SBT quantification using LC-MS/MS with application to metabolic stability evaluation.

# 1. Introduction

Bruton's tyrosine kinase (BTK) has recently become a promising drug target for many diseases, especially haematopoietic malignancies and autoimmune diseases associated with B lymphocytes. Many BTK inhibitors are currently in different stages of clinical trials. Acalabrutinib is a BTK inhibitor established by Acerta Pharma and has been approved by the FDA for adult patients with mantle cell lymphoma who have received at least one prior therapy [1].

Spebrutinib (SBT, figure 1) is an oral, bioavailable, selective inhibitor of BTK, with potential antineoplastic activity. Upon administration, SBT irreversibly and covalently binds to BTK leading to B cell receptor (BCR) signalling. It also inhibits malignancies associated with B cell proliferation. SBT, established by Avila Therapeutics (acquired by Celgene in March 2012), is currently in phase II clinical trials for rheumatoid arthritis and offers an encouraging future for the management of leukaemia and autoimmune diseases. It is also in phase I trials for chronic lymphocytic leukaemia (CLL). In 2014, Orphan Drug Designation was designated in the EU for the cure of CLL [2–5].

Estimating the bioavailability gives a valuable picture on a compound's metabolism. Drugs with rapid metabolism rates are expected to exhibit low *in vivo* bioavailability [6]. Several outcomes have indicated that SBT is a drug with a low extraction ratio and slow excretion from the human body unlike other tyrosine kinase inhibitors (TKIs) [7–9], indicating a probable high risk of dose accumulation, similar to dacomitinib [10,11]. Consequently, the SBT metabolic stability was evaluated by assessing two important parameters (*in vitro* half-life ($t_{1/2}$) and intrinsic clearance ($Cl_{int}$)) that could be used to further compute other physiological parameters (e.g. *in vivo* $t_{1/2}$, hepatic clearance and bioavailability). Metabolic stability was measured by the rate of decrease of the drug candidate when incubating with human liver microsomes (HLMs). Upon reviewing the literature, we did not find any published chromatographic methods for SBT assay. Therefore, we sought to develop an analytical method for this drug.

# 2. Experimental

## 2.1. Materials

SBT (99.95%) and naquotinib (NQT, 99.12%) were purchased from MedChem Express (USA). HLMs (M0567) along with all other chemicals and solvents were procured from Sigma-Aldrich (USA). HPLC grade water was obtained via in-house filtration using Milli-Q® reference system (Merck Millipore, MA, USA).

## 2.2. Instrumentation and conditions

Agilent RRLC 1200 was used as an HPLC system for chromatographic resolution of HLMs incubates using Agilent ZORBAX Eclipse Plus C18 column (length, 100 mm; internal diameter, 2.1 mm; and particle size, 1.8 µm). Temperature of the column was adjusted at $20 \pm 2°C$. Isocratic mobile phase was used for chromatographic resolution of SBT and internal standard (IS). The mobile phase composed of 60% aqueous part (10 mM ammonium formate in water at pH 4.2) and 40% organic part (acetonitrile). Flow rate, run time and injection volume were $0.15\,ml\,min^{-1}$, 3.5 min and 2 µl, respectively. NQT was selected as IS in SBT analysis.

A tandem mass spectrometer (Agilent 6410 QqQ; Agilent, CA, USA) with positive mode electrospray ionization (ESI) as the source interface was employed throughout the study. This ESI source used nitrogen as the drying gas with a flow rate of $12\,l\,min^{-1}$, with nitrogen being also used as the collision gas (55 psi) in the collision cell. The capillary voltage (3500 V) and ionization source temperature (350°C) were optimized. Mass Hunter software produced by Agilent (Agilent, CA, USA) was employed to control the instrument and for data collection. SBT was quantified using multiple reaction monitoring (MRM) for the mass reaction (precursor to daughter ions) from $424 \rightarrow 370$ and $424 \rightarrow 59$ for SBT, and $563 \rightarrow 463$ and $563 \rightarrow 323$ for IS (figure 2). The fragmentor voltages (FV) were 140 and 145 V with collision energy (CE) of 20 and 22 eV for SBT, and 135 and 140 V with CE of 15 and 18 eV for IS. The aforementioned transitions were chosen for SBT analysis to avoid any interfering signals from the HLMs components and increase the efficiency of the assay [12]. MRM chromatogram was detected in three segments: 0.0 to 1.0 min (to waste), 1.0 to 2.0 min (IS mass

**Figure 1.** Chemical structure of spebrutinib and naquotinib (IS).

transitions) and 2.0 to 3.5 min (SBT transitions) to avoid contamination of the mass detector with the first eluted peaks (figure 2).

## 2.3. Preparation of stock solutions and working solutions

A stock solution of SBT (1 mg ml$^{-1}$) in dimethyl sulfoxide (DMSO) was serially diluted with mobile phase to yield working solution 1 (100 µg ml$^{-1}$) and working solution 2 (10 µg ml$^{-1}$). Stock solution of IS (100 µg ml$^{-1}$) in DMSO was diluted with an appropriate amount of mobile phase to make the IS working solution (1 µg ml$^{-1}$).

## 2.4. Preparation of calibration standards and quality controls

SBT working solution 2 was combined with HLMs matrix (1 mg protein for each 1 ml phosphate buffer) to construct a calibration plot with 12 levels: 5, 10, 15, 30, 50, 80, 100, 150, 200, 300, 400 and 500 ng ml$^{-1}$. Four calibration levels were chosen; *viz.* lower limit of quantification (LLOQ, 5 ng ml$^{-1}$), low (LQC, 15 ng ml$^{-1}$), medium (MQC, 150 ng ml$^{-1}$), and high (HQC, 400 ng ml$^{-1}$) quality control solutions. IS working solution (100 µl) was added immediately before the addition of metabolic quenching agent (acetonitrile) to avoid any effect on the rate of SBT metabolism. Acetonitrile is used as a quenching agent for metabolic reaction and as precipitating agent in protein precipitation extraction procedure.

## 2.5. Extraction of spebrutinib and internal standard from human liver microsomes matrix

The protein sedimentation method using acetonitrile was employed for SBT and IS extraction as a standard method for conducting the experimental procedures for metabolic stability [13,14]. A volume of 2 ml acetonitrile was added with each 1 ml of the spiked HLMs samples and subsequently centrifuged at 14 000 r.p.m. (12 min at 4°C) to discard proteins formed in the precipitate. One millilitre of each supernatant was then gathered and filtered in a syringe filter (pore size: 0.22 µm). The filtered samples were transferred to HPLC vials. An injection volume of 2 µl was chosen for optimum enhancement of the peak shape sharpness. Two controls were prepared, as previously mentioned, using the same buffer without the HLMs matrix or NADPH. The control lacking HLMs was employed to confirm that HLMs components did not interfere at the retention times for SBT and IS. The other control lacking NADPH was exploited to assure that the change in concentration was metabolically mediated.

## 2.6. Method validation

The parameters used to validate the current LC-MS/MS assay for SBT have been described in depth in our previous articles [12,15–18]. Linearity, assay recovery, sensitivity, reproducibility, specificity, limits of quantification, (LOQ) detection (LOD) and stability were all calculated for SBT according to the US Food and Drug Administration (FDA) guidelines [19]. The least squared statistical approach was

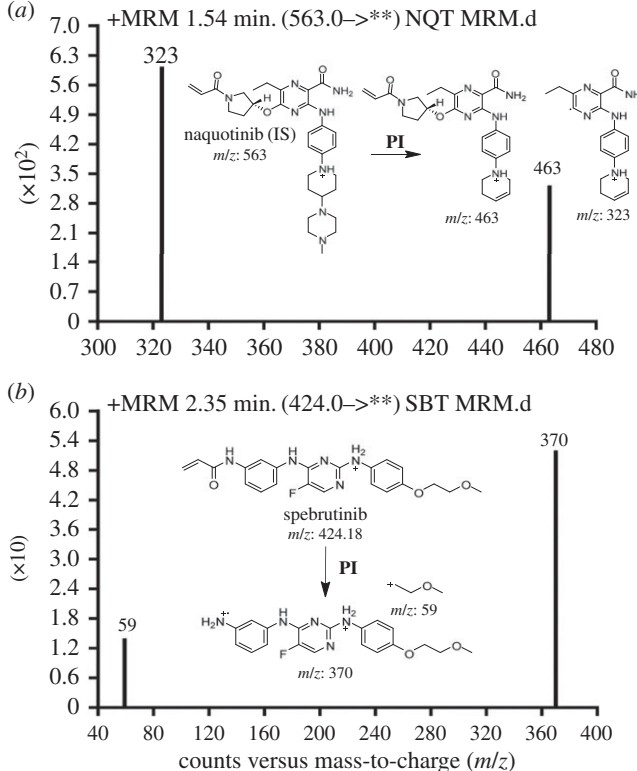

**Figure 2.** MRM mass transitions of IS (*a*) and SBT (*b*). PI, product ion.

recruited to compute the calibration plot equations ($y = ax + b$). The linear fit was confirmed using the $r^2$ value.

## 2.7. Metabolic stability evaluation of spebrutinib

Assessment of the SBT concentration with the amount remaining after incubation with HLMs was employed as the basis for evaluating the SBT metabolic stability. Briefly, incubation of 1 µM of SBT with HLMs (1 mg microsomal protein/1 ml phosphate buffer) was executed in duplicate to confirm the results using phosphate buffer (pH 7.4) that contains magnesium chloride ($MgCl_2$, 3.3 mM). The mixture was pre-incubated for 10 min in a temperature-controlled water bath (37°C). The metabolic reaction was then initiated and termination performed by respectively adding NADPH (1 mM) and 2 ml acetonitrile at specific time intervals: 0, 0.5, 2.5, 5, 10, 15, 30 and 50 min. The curve for the metabolic stability of SBT was then constructed.

# 3. Results and discussion

## 3.1. HPLC−MS/MS methodology

All parameters of the chromatographic and mass spectrometric systems were attuned to achieve the finest resolution for SBT and IS. Liquid chromatographic parameters inclusive of mobile phase (composition and pH) and stationary phase were adjusted to accomplish optimum resolution with a fast run time. For the aqueous portion (10 mM ammonium formate) of the elution phase, pH was optimized to 4.2 using formic acid. With a higher pH, retention time increased and peak front tailing was observed. The ratio between the aqueous (10 mM ammonium formate) and organic part (Acetonitrile) was set to 60 : 40. This is because an increase in acetonitrile resulted in poor resolution as well as overlapping chromatographic peaks, while a decrease in acetonitrile increased the run time. We then proceeded to test different types of columns. SBT and IS were retained and good results were achieved using a reverse-phase C18 column. The time for SBT and IS elution was 3.5 min and good separation was achieved. In addition, we did not find any carry-over influence in blank HLMs sample.

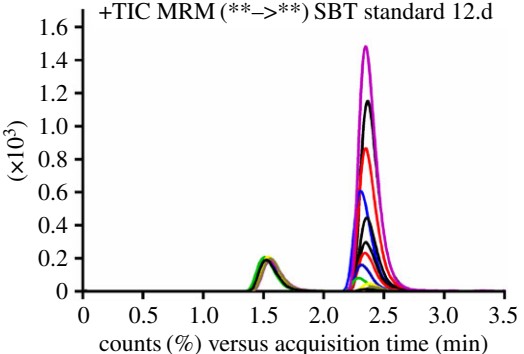

**Figure 3.** Overlaid multiple reaction monitoring chromatograms of 12 calibration levels of SBT with IS (100 ng ml$^{-1}$).

NQT was carefully chosen as the IS for the SBT analysis, because the same method of extraction from HLMs matrix could be applied for both SBT and NQT, and their recoveries were $101.9 \pm 5.8\%$ and $98.7 \pm 0.7\%$, respectively. The chromatographic peak of NQT (1.5 min) is near the retention time of SBT (2.4 min). That supports the proposed method objective of being fast (3.5 min). Both NQT and SBT are TKIs, and are not clinically administered together to any individual patient simultaneously. Therefore, the described method herein may be beneficial for various clinical applications (e.g. therapeutic drug monitoring or pharmacokinetics) for patients under SBT treatment.

MRM was exploited for the SBT assay to increase the sensitivity of the current method and eliminate any probable interfering signals from the HLMs matrix (figure 2). Flow injection analysis was used for optimization of ionization and fragmentation parameters including fragmentor voltage and collision energy to get the most intense fragment ions from SBT and IS. Figure 3 shows the QC standards for SBT as overlaid MRM chromatograms.

## 3.2. Validation of the LC-MS/MS method

### 3.2.1. Specificity

We observed a good resolution of the chromatographic peaks for SBT and IS (figure 4). In addition, a blank HLMs matrix revealed the absence of peaks in the retention times for the analyte, revealing the specificity of the developed method. No carry-over influence of SBT and IS in the MRM chromatograms was observed.

### 3.2.2. Sensitivity and linearity

The linearity concentration range and the determined correlation coefficient ($r^2$) for the current method were 5–500 ng ml$^{-1}$ and $\geq 0.9999$, respectively, with $Y = 1.4518x - 5.0624$ as the observed SBT regression equation obtained from calibration plot. LOD and LOQ values were figured to be 0.39 and 1.19 ng ml$^{-1}$, respectively, and the LLQC peak exhibited good peak shape with high signal to noise (S/N) ratio, confirming the method sensitivity (figure 5).

Values of less than 3.96% were calculated as the RSD for six replicates for each standard were in HLMs (table 1). Reverse calculations of the 12 SBT calibration levels in HLMs confirmed the effectiveness of the described assay.

### 3.2.3. Precision and accuracy

The precision and accuracy values acquired from intra-day and inter-day assessments were found to range from 0.71% to 4.78%, and from $-1.41$ to 12.44, respectively (table 2). Average SBT recovery was $101.9 \pm 5.8\%$ in the HLMs. Those estimated values were deemed acceptable relying on the FDA guidelines (table 2).

### 3.2.4. Influences of the matrix and extraction efficiency

The SBT and IS recoveries in the HLMs matrix were $101.9 \pm 5.8\%$ and $98.7 \pm 0.7\%$, respectively (table 3). Based on the subsequent analysis, we did not confirm any matrix effect on SBT or IS when the two HLMs

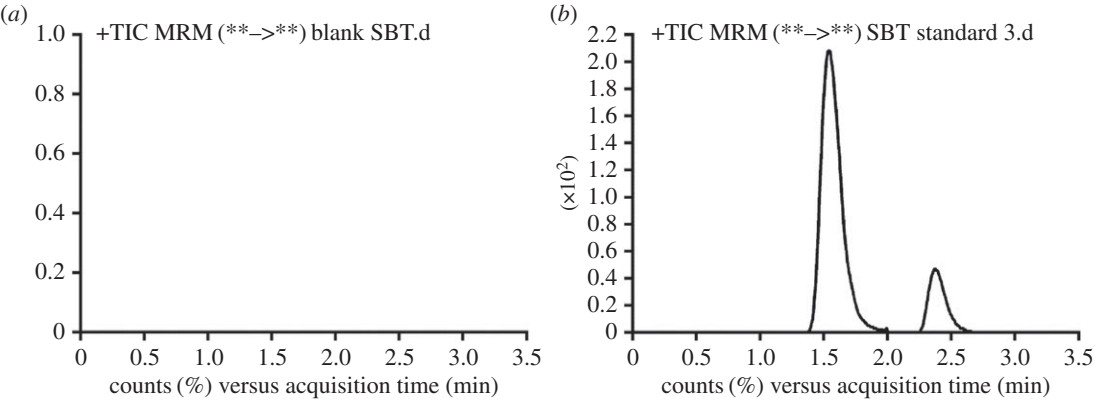

**Figure 4.** Multiple reaction monitoring chromatograms for (*a*) blank human liver microsomes and (*b*) the lower quality control of SBT (15 ng ml$^{-1}$).

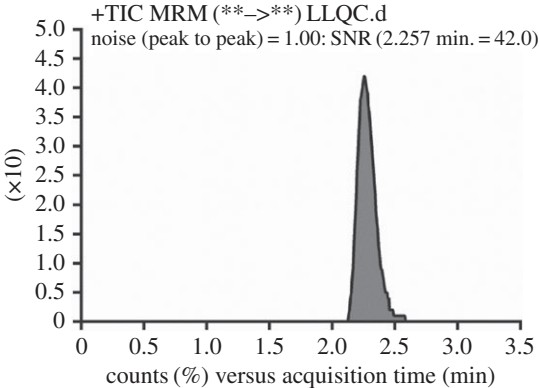

**Figure 5.** The lower limit of quantification for SBT exhibited high signal to noise ratio.

samples spiked with the SBT LQC (15 ng ml$^{-1}$) and IS (100 ng ml$^{-1}$) were analysed; these samples were labelled as Set 1. The mobile phase solution was added as a substitute to the HLMs matrix to prepare Set 2. The influence exerted by the matrix was determined using the following equations:

$$\text{matrix effect of SBT} = \text{mean peak area ratio } \frac{\text{Set 1}}{\text{Set 2}} \times 100 \quad \text{and}$$

$$\text{matrix effect of IS} = \text{mean peak area ratio } \frac{\text{Set 1}}{\text{Set 2}} \times 100.$$

The HLMs consisting of SBT and IS exhibited matrix effects of $101.9 \pm 5.8\%$ and $98.7 \pm 0.7\%$, respectively. The internal standard matrix effect (IS normalized MF) was estimated to be 1.02 via the following formula, and was found to be within an adequate range [8]:

$$\text{IS normalized MF} = \frac{\text{matrix effect of SBT}}{\text{matrix effect of IS}}.$$

These results demonstrate that there were no observed influences of the HLMs on the SBT and IS ionization.

### 3.2.5. Stability

SBT stability in HLMs matrix was tested under all laboratory conditions that might have been subjected during experimental analysis. SBT exhibited good stability in HLMs matrix after storage at $-20°C$ for 28 days as stability values were ranged from 96.42% to 107.02%. SBT stability data is summarized in table 4. There was no noticeable degradation of analytes under the examined conditions indicating that STB exhibited good stability in all laboratory conditions.

**Table 1.** Back-calculation of the calibration levels of SBT in the human liver microsome matrix. LLOQ, lower limit of quantification; LQC, lower quality control; MQC, medium quality control; HQC, high quality control.

| SBT calibration level in ng ml$^{-1}$ | mean[a] | s.d. | RSD (%) | accuracy (%) |
|---|---|---|---|---|
| 5 (LLOQ) | 5.62 | 0.22 | 3.96 | 12.44 |
| 10 | 10.37 | 0.21 | 2.02 | 3.73 |
| 30 | 28.06 | 0.87 | 3.08 | −6.48 |
| 50 | 51.25 | 1.21 | 2.37 | 2.49 |
| 80 | 79.49 | 1.47 | 1.85 | −0.64 |
| 100 | 99.35 | 1.35 | 1.36 | −0.65 |
| 200 | 200.23 | 2.75 | 1.37 | 0.11 |
| 300 | 299.36 | 2.31 | 0.77 | −0.21 |
| 500 | 497.03 | 3.49 | 0.70 | −0.59 |
| quality controls | | | | |
| 15 (LQC) | 14.79 | 0.18 | 1.25 | −1.41 |
| 150 (MQC) | 149.36 | 1.82 | 1.22 | −0.43 |
| 400 (HQC) | 399.24 | 3.05 | 0.76 | −0.19 |

[a]Average of six repeats.

**Table 2.** Intra-day and inter-day assay results for the developed method.

| QC level | LLQC (5 ng ml$^{-1}$) | | LQC (15 ng ml$^{-1}$) | | MQC (150 ng ml$^{-1}$) | | HQC (400 ng ml$^{-1}$) | |
|---|---|---|---|---|---|---|---|---|
| assay | intra-day[a] | inter-day[b] | intra-day | inter-day | intra-day | inter-day | intra-day | inter-day |
| mean | 5.62 | 5.52 | 14.79 | 14.83 | 149.36 | 149.08 | 399.24 | 399.71 |
| s.d. | 0.22 | 0.26 | 0.18 | 0.23 | 1.82 | 1.87 | 3.05 | 2.83 |
| % RSD | 3.96 | 4.78 | 1.25 | 1.53 | 1.22 | 1.25 | 0.76 | 0.71 |
| % accuracy | 12.44 | 10.46 | −1.41 | −1.15 | −0.43 | −0.62 | −0.19 | −0.07 |

[a]Average of 12 repeats in 1 day.
[b]Average of six repeats in 3 days.

**Table 3.** Recovery of the SBT samples in HLMs matrix.

| QC levels | HLM matrix | | | |
|---|---|---|---|---|
| | LLOQ (5 ng ml$^{-1}$) | LQC (15 ng ml$^{-1}$) | MQC (150 ng ml$^{-1}$) | HQC (400 ng ml$^{-1}$) |
| mean | 5.62 | 14.79 | 149.36 | 399.24 |
| s.d. | 0.22 | 0.18 | 1.82 | 3.05 |
| precision (RSD %) | 3.96 | 1.25 | 1.22 | 0.76 |
| recovery (%) | 110.60 | 99.10 | 98.87 | 99.01 |
| SBT recovery | 101.9 ± 5.8% | | | |

## 3.3. SBT metabolic stability investigation

The concentration of SBT in the HLMs was figured by means of a calibration graph regression equation. The SBT metabolic stability graph was established by tracing the ln of the remaining % of SBT versus the time intervals (figure 6).

**Table 4.** Stability of SBT under different storage conditions.

| nominal concentrations of SBT in ng ml$^{-1}$ | mean[a] | s.d. | RSD (%) | accuracy (%) |
|---|---|---|---|---|
| room temperature for 8 h | | | | |
| 5 | 5.28 | 0.13 | 2.55 | 105.50 |
| 15 | 14.85 | 0.28 | 1.89 | 99.02 |
| 150 | 149.13 | 2.69 | 1.81 | 99.42 |
| 400 | 400.40 | 2.68 | 0.67 | 100.10 |
| three freeze–thaw cycles | | | | |
| 5 | 5.35 | 0.15 | 2.90 | 107.02 |
| 15 | 14.96 | 0.15 | 1.01 | 99.72 |
| 150 | 148.69 | 2.25 | 1.51 | 99.13 |
| 400 | 400.04 | 2.40 | 0.60 | 100.01 |
| stored at 4°C for 24 h | | | | |
| 5 | 5.30 | 0.12 | 2.32 | 106.08 |
| 15 | 14.80 | 0.18 | 1.22 | 98.68 |
| 150 | 149.36 | 1.71 | 1.14 | 99.57 |
| 400 | 399.35 | 3.01 | 0.75 | 99.84 |
| stored at −20°C for 30 days | | | | |
| 5 | 5.19 | 0.31 | 6.06 | 103.78 |
| 15 | 14.46 | 0.75 | 5.18 | 96.42 |
| 150 | 148.61 | 3.55 | 2.39 | 99.07 |
| 400 | 397.45 | 6.20 | 1.56 | 99.36 |

[a]Average of six replicates.

The initial segment of the curve was linear and the obtained regression equation ($y = -0.0084x + 4.5939$; $r^2 = 0.9397$) was applied to figure the *in vitro* $t_{1/2}$ [13,20] (table 5).

The following equations were also used:

$$\textit{in vitro } t_{1/2} = \frac{\ln 2}{\text{slope}}$$

(slope = 0.0044)

$$\textit{in vitro } t_{1/2} = \frac{\ln 2}{0.0084}$$
$$\textit{in vitro } t_{1/2} = 82.518 \text{ min.}$$

Computing the inherent clearance of SBT was executed via the use of the *in vitro* $t_{1/2}$ method [21,22] using the following equation:

$$Cl_{int,app} = \frac{\ln 2}{\textit{in vitro } t_{1/2}} \cdot \frac{\mu l \text{ incubation}}{\text{mg protein}},$$

$$Cl_{int,app} = \frac{0.693}{82.52} \cdot \frac{1000}{1}$$

and

$$Cl_{int,app} = 8.4 \ \mu l \min^{-1} mg^{-1}.$$

These outcomes demonstrate that the SBT metabolic stability was denoted by a very low $Cl_{int}$ (8.4 μl/min mg$^{-1}$) and very long *in vitro* $t_{1/2}$ value (82.52 min). In addition, SBT exhibited a low metabolic clearance from the human body via the liver and thus, might be accumulated inside the body, similar to dacomitinib.

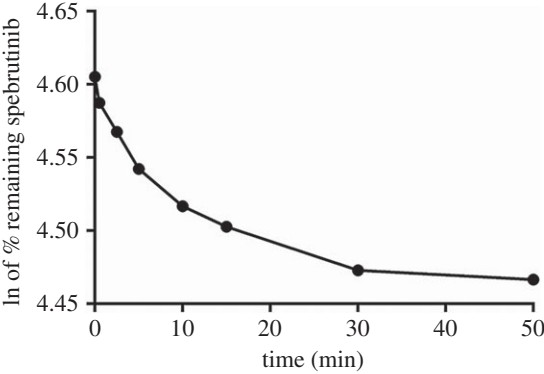

**Figure 6.** Curve representing the metabolic stability of SBT in human liver microsomes.

**Table 5.** Parameters for the metabolic stability of SBT.

| time (min) | conc. (ng ml$^{-1}$) | LN of % SBT remaining | parameter | value |
|---|---|---|---|---|
| 0.0 | 423.0 | 4.61 | linear part regression equation | $y = -0.0084x + 4.5939$ |
| 0.5 | 415.5 | 4.59 | | |
| 2.5 | 407.4 | 4.57 | $r^2$ | 0.9397 |
| 5.0 | 397.2 | 4.54 | | |
| 10.0 | 387.2 | 4.52 | slope | 0.008 |
| 15.0 | 381.8 | 4.50 | | |
| 30.0 | 370.6 | 4.47 | $t_{1/2}$ | 82.52 min |
| 50.0 | 368.3 | 4.47 | $Cl_{int}$ | 8.4 $\mu$l min$^{-1}$ mg$^{-1}$ |

# 4. Conclusion

A highly sensitive LC-MS/MS method to assay SBT was developed and validated. The proposed method was fast and accurate and displayed high recovery and good sensitivity. Additionally, the described method can be claimed a green chemistry approach with the low volume of organic solvent (acetonitrile) being consumed during such assay. This approach was exploited to study the SBT metabolic stability in HLMs yielding two key parameters: *in vitro* $t_{1/2}$ and $Cl_{int}$. The outcomes indicated that SBT might be accumulating in the body and slowly eliminated by the liver. This encourages further research into the pharmacokinetics of SBT with close monitoring of SBT level during treatment.

# Abbreviations

BTK, Bruton's tyrosine kinase; CE, collision energy; $Cl_{int}$, intrinsic clearance; CLL, chronic lymphocytic leukaemia; DMSO, dimethyl sulfoxide; ESI, electrospray ionization; FV, fragmentor voltages; HLMs, human liver microsome; IS, internal standard; LC-MS/MS, liquid chromatography tandem mass spectrometry; MRM, multiple reaction monitoring; NQT, naquotinib; SBT, spebrutinib; $t_{1/2}$, half-life; TKIs, tyrosine kinase inhibitors.

Ethics. The heading does not apply as HLMs were purchased from Sigma Aldrich company and used in *in vitro* metabolic investigation for SBT.

Data accessibility. The data supporting the results in this article can be accessed from the Dryad Digital Repository: https://doi.org/10.5061/dryad.5m09t8k [23].

Authors' contributions. M.W.A. and A.S.A. were participated in research designing and experimental work supervision. A.S.A., M.W.A. and A.A.K. performed the adjustment and method validation studies. M.W.A. wrote the first draft of the manuscript. M.W.A. and N.S.A. prepared the final revised draft of the manuscript and replied to reviewers comments. All authors revised and approved the final form of the manuscript.

Competing interests. We declare we have no competing interests.
Funding. The authors would like to extend their sincere appreciation to the Deanship of Scientific Research at the King Saud University for funding this work through the Research Group Project no. RG-1435-025.

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
