## [Reviewer comments · Royal Society Open Science]

Review History

RSOS-190434.R0 (Original submission)

Review form: Reviewer 1 (Nageswara Rao Pilli)

Is the manuscript scientifically sound in its present form?

Yes

Are the interpretations and conclusions justified by the results?

Yes

Is the language acceptable?

Yes

Is it clear how to access all supporting data?

Yes

Do you have any ethical concerns with this paper?

No

Have you any concerns about statistical analyses in this paper?

No

Recommendation?

Accept with minor revision (please list in comments)

Comments to the Author(s)

The authors developed a LC-MS assay for the determination of Spebrutinib in human liver microsomes. This method is properly developed and having enough scientific merits to publish. But Authors should have been taken more care in manuscript preparation and data presentation. Many formatting errors in this manuscript. I request Authors to revise the manuscript as per the comments below:

1. Delete the word "Accurately" in title. Also, LC-MS should be LC-MS/MS. Avoid abbreviations in Abstract. Provide list all abbreviations used. Abbreviate for the first time and used abbreviation thereafter (example Internal standard, IS; Page 4 lane 21-22; in Table 1).
2. In abstract Lane 15: it should be 5-500 ng/mL.
3. Please cite all figures and tables in the text. Avoid jumping of the sentence Example: Introduction Page 3 Lane 32-33. Authors explained about the available literature. In next lanes again explained about the drug profile. The order should be explained about the drug profile, PK, PD etc, followed by literature review and highlights of the proposed method or details of the proposed method.
4. Section 2.2, please modify as "Instrumentation and conditions". This section should explain only about the optimized chromatographic and mass spectrometric conditions of the proposed method. Please delete details about how the IS was selected. This should be explained in the results and discussion section. Also, the details provided in Page 4, Lane 38-54, is almost like details in Table 1. Its duplication of the data. Hence, request to provide all LC-MS/MS conditions in text or Table, either one.
5. Provide Scheme 1 as Supplementary Data or with Figure 2.
6. Section 2.3 change to "preparation of Stock Solutions and Working Solutions". Section 2.4 "Preparation of Calibration standards and Quality Controls". Replace ACN with acetonitrile in entire manuscript. Page 6, Lane 32-38 "A calibration graph..." delete, not necessary, its duplication.
7. It was not clear when the IS was added to samples. As per the Lane 21-22 of Page 6, the IS was added to the samples after processing. In Bioanalysis, the use of Internal standard is to avoid the sample processing and chromatography related errors (others too). But in the present method the IS was added after the sample precipitation, means after complete extraction of analytes, where there is a scope of extraction errors or possible loss. Reason?
8. Page 6, lane 51-53; Assay recovery...Please delete sentence. This section should explain only about how the validation was carried out and not the results.
9. Results and discussion: Explain how mass spectrometry conditions were optimized, Chromatography, Extraction conditions optimization and reason for selecting NQT as IS.
10. As part of Bioanalytical method validation, analyte stability in matrix (at room temperature, Freeze & thaw, and upon long-term storage etc) as well as in neat sample (eg stock solution stability) and processed samples stability should be established.
11. What is the C18 column brand used.

Review form: Reviewer 2**Is the manuscript scientifically sound in its present form?**

Yes

Are the interpretations and conclusions justified by the results?

Yes

Is the language acceptable?

Yes

Is it clear how to access all supporting data?

Yes

Do you have any ethical concerns with this paper?

Yes

Have you any concerns about statistical analyses in this paper?

No

Recommendation?

Accept with minor revision (please list in comments)

Comments to the Author(s)

Title: Acceptable, The name of the detection technique that is LC-MS should be changed into LC-MS/MS.

Abstract: Acceptable

Abbreviations should not be used in the abstract. Spare this abbreviation to first appearance in the text. Abbreviations should be provided in a separate section.

Keywords: Acceptable

"First letter of each keyword shouldn't be capitalized. Add intrinsic clearance as a keyword.

Introduction:

Separate paragraph should be included explaining the importance of metabolic stability investigation.

Methods:

1- NQT used as IS, so it should be mentioned in the manuscript as IS.

2- Table 1 should be removed and all chromatographic conditions should be mentioned in a text.

3- Assay recovery should be removed from the methodology and should be mentioned in results.

4- Method of extraction of SBT from HLMs matrix should be mentioned in details a separate section.

Results:

1- Assay recovery should be included in a separate section.

2- Figures: Acceptable but author should supply a high resolution figures as some figures are vague.

Conclusions: It should be concise as it contains duplication from the manuscript sentences.

References:

Journal names should be abbreviated.

Tables: Acceptable. Table 1 should be removed

Decision letter (RSOS-190434.R0)

15-Apr-2019

Dear Dr Attwa:

Title: A highly sensitive LC-MS method to accurately determine novel BTK inhibitor spebrutinib: Application to metabolic stability evaluation
Manuscript ID: RSOS-190434

Thank you for submitting the above manuscript to Royal Society Open Science. On behalf of the Editors and the Royal Society of Chemistry, I am pleased to inform you that your manuscript will be accepted for publication in Royal Society Open Science subject to minor revision in accordance with the referee suggestions. Please find the reviewers' comments at the end of this email.

The reviewers and handling editors have recommended publication, but also suggest some minor revisions to your manuscript. Therefore, I invite you to respond to the comments and revise your manuscript.

Please also include the following statements alongside the other end statements. As we cannot publish your manuscript without these end statements included, if you feel that a given heading is not relevant to your paper, please nevertheless include the heading and explicitly state that it is not relevant to your work. We have included a screenshot example of the end statements for reference.

- Ethics statement

Please clarify whether you received ethical approval from a local ethics committee to carry out your study. If so please include details of this, including the name of the committee that gave consent in a Research Ethics section after your main text. Please also clarify whether you received informed consent for the participants to participate in the study and state this in your Research Ethics section.

OR

Please clarify whether you obtained the necessary licences and approvals from your institutional animal ethics committee before conducting your research. Please provide details of these licences and approvals in an Animal Ethics section after your main text.

OR

Please clarify whether you obtained the appropriate permissions and licences to conduct the fieldwork detailed in your study. Please provide details of these in your methods section.

- Data accessibility

It is a condition of publication that you make available the data and research materials supporting the results in the article. Datasets should be deposited in an appropriate publicly available repository and details of the associated accession number, link or DOI to the datasets must be included in the Data Accessibility section of the article (<http://royalsocietypublishing.org/instructions-authors#question17>). Reference(s) to datasets should also be included in the reference list of the article with DOIs (where available).

Please include a Data Availability section after your main text stating where supporting data are available from, or where they will be made available should your article be accepted for publication.

<http://datadryad.org/submit?journalID=RSOS&manu=RSOS-190434>

- **Competing interests**

Please include a Competing Interests section after your main text declaring any financial or non-financial competing interests. If you have no competing interests please state 'I/we have no competing interests.'

- **Authors' contributions**

Please include an Authors' Contributions section at the end of your main text detailing the contribution of each author. All authors should have read and approved the manuscript before submission and this should be stated in the Authors' Contributions section.

The list of Authors should meet all of the following criteria; 1) substantial contributions to conception and design, or acquisition of data, or analysis and interpretation of data; 2) drafting the article or revising it critically for important intellectual content; and 3) final approval of the version to be published.

- **Acknowledgements**

- **Funding statement**

Please include a funding section after your main text which lists the source of funding for each author.

Because the schedule for publication is very tight, it is a condition of publication that you submit the revised version of your manuscript before 24-Apr-2019. Please note that the revision deadline will expire at 00.00am on this date. If you do not think you will be able to meet this date please let me know immediately.

Best wishes,
Dr Laura Smith
Publishing Editor, Journals

Reviewer comments to Author:
Reviewer: 1

Comments to the Author(s)

The authors developed a LC-MS assay for the determination of Spebrutinib in human liver microsomes. This method is properly developed and having enough scientific merits to publish. But Authors should have been taken more care in manuscript preparation and data presentation.

Many formatting errors in this manuscript. I request Authors to revise the manuscript as per the comments below:

1. Delete the word "Accurately" in title. Also, LC-MS should be LC-MS/MS. Avoid abbreviations in Abstract. Provide list all abbreviations used. Abbreviate for the first time and used abbreviation thereafter (example Internal standard, IS; Page 4 lane 21-22; in Table 1).
2. In abstract Lane 15: it should be 5-500 ng/mL.
3. Please cite all figures and tables in the text. Avoid jumping of the sentence Example: Introduction Page 3 Lane 32-33. Authors explained about the available literature. In next lanes again explained about the drug profile. The order should be explained about the drug profile, PK, PD etc, followed by literature review and highlights of the proposed method or details of the proposed method.
4. Section 2.2, please modify as "Instrumentation and conditions". This section should explain only about the optimized chromatographic and mass spectrometric conditions of the proposed method. Please delete details about how the IS was selected. This should be explained in the results and discussion section. Also, the details provided in Page 4, Lane 38-54, is almost like details in Table 1. Its duplication of the data. Hence, request to provide all LC-MS/MS conditions in text or Table, either one.
5. Provide Scheme 1 as Supplementary Data or with Figure 2.
6. Section 2.3 change to "preparation of Stock Solutions and Working Solutions". Section 2.4 "Preparation of Calibration standards and Quality Controls". Replace ACN with acetonitrile in entire manuscript. Page 6, Lane 32-38 "A calibration graph..." delete, not necessary, its duplication.
7. It was not clear when the IS was added to samples. As per the Lane 21-22 of Page 6, the IS was added to the samples after processing. In Bioanalysis, the use of Internal standard is to avoid the sample processing and chromatography related errors (others too). But in the present method the IS was added after the sample precipitation, means after complete extraction of analytes, where there is a scope of extraction errors or possible loss. Reason?
8. Page 6, lane 51-53; Assay recovery...Please delete sentence. This section should explain only about how the validation was carried out and not the results.
9. Results and discussion: Explain how mass spectrometry conditions were optimized, Chromatography, Extraction conditions optimization and reason for selecting NQT as IS.
10. As part of Bioanalytical method validation, analyte stability in matrix (at room temperature, Freeze & thaw, and upon long-term storage etc) as well as in neat sample (eg stock solution stability) and processed samples stability should be established.
11. What is the C18 column brand used.

Reviewer: 2

Comments to the Author(s)

Title: Acceptable, The name of the detection technique that is LC-MS should be changed into LC-MS/MS.

Abstract: Acceptable

Abbreviations should not be used in the abstract. Spare this abbreviation to first appearance in the text. Abbreviations should be provided in a separate section.

Keywords: Acceptable

"First letter of each keyword shouldn't be capitalized. Add intrinsic clearance as a keyword.

Introduction:

Separate paragraph should be included explaining the importance of metabolic stability investigation.

Methods:

- 1- NQT used as IS, so it should be mentioned in the manuscript as IS.
- 2- Table 1 should be removed and all chromatographic conditions should be mentioned in a text.
- 3- Assay recovery should be removed from the methodology and should be mentioned in results.
- 4- Method of extraction of SBT from HLMs matrix should be mentioned in details a separate section.

Results:

- 1- Assay recovery should be included in a separate section.
- 2- Figures: Acceptable but author should supply a high resolution figures as some figures are vague.

Conclusions: It should be concise as it contains duplication from the manuscript sentences.

References:

Journal names should be abbreviated.

Tables: Acceptable. Table 1 should be removed

Author's Response to Decision Letter for (RSOS-190434.R0)

See Appendix A.

Decision letter (RSOS-190434.R1)

30-Apr-2019

Dear Dr Attwa:

Title: A highly sensitive LC-MS/MS method to determine novel BTK inhibitor spebrutinib:
Application to metabolic stability evaluation
Manuscript ID: RSOS-190434.R1

It is a pleasure to accept your manuscript in its current form for publication in Royal Society Open Science. The chemistry content of Royal Society Open Science is published in collaboration with the Royal Society of Chemistry.

Royal Society of Chemistry
Thomas Graham House

Science Park, Milton Road
Cambridge, CB4 0WF
Royal Society Open Science - Chemistry Editorial Office

RSC Associate Editor
Comments to the Author:
(There are no comments.)

Reviewer(s)' Comments to Author:

Appendix A

[17th April 2019]

Dr Laura Smith
Publishing Editor

Royal Society Open Science Journal

Dear Editor,

We wish to re-submit the revised version of the manuscript titled “**A highly sensitive LC-MS/MS method to determine novel BTK inhibitor spebrutinib: Application to metabolic stability evaluation.**” Manuscript ID RSOS-190434.

We thank the reviewers for their thoughtful suggestions and valuable comments regarding our work. The manuscript has benefited from these insightful suggestions. I look forward to working with you and the reviewers to move this manuscript closer to publication in the *Royal Society Open Science Journal*.

The manuscript has been rechecked, and the necessary changes have been made in accordance with the reviewers' suggestions. The point-by-point responses to the comments raised by the respective reviewers have been prepared and attached below, and all changes have been highlighted in the revised manuscript.

Thank you for your consideration I look forward to hearing from you.

Sincerely,

Mohamed W. Attwa
Department of Pharmaceutical Chemistry
College of Pharmacy
King Saud University
P.O. Box 2457 Riyadh, 11451, Saudi Arabia
Tel.: +966 1146 70237
Fax: +966 1146 76 220
E-mail: mzeidan@ksu.edu.sa

REVIEWER REPORT(S):

Reviewer 1

We appreciate the reviewer for these pertinent comments and we have addressed them all in the revised manuscript.

Comments to the Author(s)

The authors developed a LC-MS assay for the determination of Spebrutinib in human liver microsomes. This method is properly developed and having enough scientific merits to publish. But Authors should have been taken more care in manuscript preparation and data presentation. Many formatting errors in this manuscript. I request Authors to revise the manuscript as per the comments below:

1. Delete the word "Accurately" in title. Also, LC-MS should be LC-MS/MS. Avoid abbreviations in Abstract. Provide list all abbreviations used. Abbreviate for the first time and used abbreviation there after (example Internal standard, IS; Page 4 lane 21-22; in Table 1).
 - a- The title was corrected as requested to read: A highly sensitive LC-MS/MS method to determine novel BTK inhibitor spebrutinib: Application to metabolic stability evaluation.
 - b- The list of abbreviations was provided.
 - c- The manuscript was revised and abbreviations were adjusted.
2. In abstract Lane 15: it should be 5-500 ng/mL.

We corrected this as requested.

3. Please cite all figures and tables in the text. Avoid jumping of the sentence Example: Introduction Page 3 Lane 32-33. Authors explained about the available literature. In next lanes again explained about the drug profile. The order should be explained about the drug profile, PK, PD etc, followed by literature review and highlights of the proposed method or details of the proposed method.

We rearranged the paragraph as requested and we cited all figures and tables.

4. Section 2.2, please modify as "Instrumentation and conditions". This section should explain only about the optimized chromatographic and mass spectrometric conditions of the proposed method. Please delete details about how the IS was selected. This should be explained in the results and discussion section. Also, the details provided in Page 4, Lane 38-54, is almost like details in Table 1. Its duplication of the data. Hence, request to provide all LC-MS/MS conditions in text or Table, either one.
 - a- The title of the section was modified as requested.
 - b- Table 1 was removed.
 - c- The section was revised and rearranged as requested.

5. Provide Scheme 1 as Supplementary Data or with Figure 2.

We moved scheme 1 to figure 2 as requested.

6. Section 2.3 change to “preparation of Stock Solutions and Working Solutions”. Section 2.4 “Preparation of Calibration standards and Quality Controls”. Replace ACN with acetonitrile in entire manuscript. Page 6, Lane 32-38 “A calibration graph...” delete, not necessary, its duplication.

We did as requested.

7. It was not clear when the IS was added to samples. As per the Lane 21-22 of Page 6, the IS was added to the samples after processing. In Bioanalysis, the use of Internal standard is to avoid the sample processing and chromatography related errors (others too). But in the present method the IS was added after the sample precipitation, means after complete extraction of analytes, where there is a scope of extraction errors or possible loss. Reason?

a- We added IS just before the addition of acetonitrile to avoid any effect of IS on the metabolism of SBT. We clarified this in the updated manuscript to read:

IS working solution (100 µL) was added immediately before the addition of metabolic quenching agent (acetonitrile) to avoid any effect on the rate of SBT metabolism. Acetonitrile is used as a quenching agent for metabolic reaction and as precipitating agent in protein precipitation extraction procedure.

b- We calculated the extraction recovery for IS as indicated in the text and we measured the matrix effect on IS ionization

8. Page 6, lane 51-53; Assay recovery...Please delete sentence. This section should explain only about how the validation was carried out and not the results.

We did as requested and rearranged this paragraph.

9. Results and discussion: Explain how mass spectrometry conditions were optimized, Chromatography, Extraction conditions optimization and reason for selecting NQT as IS.

We rearranged the whole section including requested information.

10. As part of Bioanalytical method validation, analyte stability in matrix (at room temperature, Freeze & thaw, and upon long-term storage etc) as well as in neat sample (eg stock solution stability) and processed samples stability should be established.

These experiments were done before during validation but we thought it is not necessary as HLMs is not a biological fluid as human plasma that can be taken from patient and stored. HLMs mixed with phosphate buffer just before the experiment. We had the data and we update the required details in the manuscript.

11. What is the C18 column brand used.

Agilent ZORBAX Eclipse Plus C18 column (length, 100 mm; internal diameter, 2.1 mm; and particle size, 1.8 µm).

We updated this in the manuscript at section 2.2. Instrumentation and conditions

Reviewer 2

We appreciate the reviewer for these pertinent comments and we have addressed them all in the revised manuscript.

Comments to the Author(s)

- 1- Title: Acceptable, The name of the detection technique that is LC-MS should be changed into LC-MS/MS.

We changed the title as requested.

- 2- Abstract:
 - a- Acceptable Abbreviations should not be used in the abstract. Spare this abbreviation to first appearance in the text.

We did as requested.

- b- Abbreviations should be provided in a separate section.

We make abbreviations in a separate section after the title.

- 3- Keywords:

- a- Acceptable "First letter of each keyword shouldn't be capitalized.

We changed the text as requested.

- b- Add intrinsic clearance as a keyword.

We added it as requested.

- 4- Introduction:

Separate paragraph should be included explaining the importance of metabolic stability investigation.

We rearranged the introduction as requested.

- 5- Methods:

- a- NQT used as IS, so it should be mentioned in the manuscript as IS.

We did as requested.

- b- Table 1 should be removed and all chromatographic conditions should be mentioned in a text.

We removed table 1 and we replaced it with text.

- c- Assay recovery should be removed from the methodology and should be mentioned in results.

We did as requested.

- d- Method of extraction of SBT from HLMS matrix should be mentioned in details a separate section.

We did as requested.

- 6- Results:

- a- Assay recovery should be included in a separate section.

We did as requested.

b- Figures: Acceptable but author should supply a high resolution figures as some figures are vague.

We did as requested.

7- Conclusions: It should be concise as it contains duplication from the manuscript sentences.

We did as requested.

8- References: Journal names should be abbreviated. Tables: Acceptable. Table 1 should be removed.

No need to abbreviate journal names as this is not in the requested style by the Royal Society Open Science Journal (Vancouver style).